# Structural Basis for Agonistic Activity and Selectivity toward Melatonin Receptors *h*MT1 and *h*MT2

**DOI:** 10.3390/ijms24032863

**Published:** 2023-02-02

**Authors:** Mattia Cantarini, Dario Rusciano, Rosario Amato, Alessio Canovai, Maurizio Cammalleri, Massimo Dal Monte, Cristina Minnelli, Emiliano Laudadio, Giovanna Mobbili, Giorgia Giorgini, Roberta Galeazzi

**Affiliations:** 1Department DISVA, Università Politecnica delle Marche, Via Brecce Bianche, 60131 Ancona, Italy; 2Fidia Pharma Group, Research Center, 95100 Catania, Italy; 3Department of Biology, University of Pisa, 56127 Pisa, Italy; 4Department SIMAU, Università Politecnica delle Marche, Via Brecce Bianche, 60131 Ancona, Italy

**Keywords:** melatonin receptors, glaucoma, melatonergic agonists, molecular docking, molecular dynamics, drug design

## Abstract

Glaucoma, a major ocular neuropathy originating from a progressive degeneration of retinal ganglion cells, is often associated with increased intraocular pressure (IOP). Daily IOP fluctuations are physiologically influenced by the antioxidant and signaling activities of melatonin. This endogenous modulator has limited employment in treating altered IOP disorders due to its low stability and bioavailability. The search for low-toxic compounds as potential melatonin agonists with higher stability and bioavailability than melatonin itself could start only from knowing the molecular basis of melatonergic activity. Thus, using a computational approach, we studied the melatonin binding toward its natural macromolecular targets, namely melatonin receptors 1 (MT1) and 2 (MT2), both involved in IOP signaling regulation. Besides, agomelatine, a melatonin-derivative agonist and, at the same time, an atypical antidepressant, was also included in the study due to its powerful IOP-lowering effects. For both ligands, we evaluated both stability and ligand positioning inside the orthosteric site of MTs, mapping the main molecular interactions responsible for receptor activation. Affinity values in terms of free binding energy (ΔG_bind_) were calculated for the selected poses of the chosen compounds after stabilization through a dynamic molecular docking protocol. The results were compared with experimental in vivo effects, showing a higher potency and more durable effect for agomelatine with respect to melatonin, which could be ascribed both to its higher affinity for hMT2 and to its additional activity as an antagonist for the serotonin receptor 5-HT2c, in agreement with the in silico results.

## 1. Introduction

Glaucoma is a chronic, long-term disease that can be classified within the ocular neuropathies characterized by progressive degeneration of retinal ganglion cells [1]. Nowadays, glaucoma pharmacological and surgical treatments are focused on IOP control in an attempt to delay its progression. One of the main factors contributing to glaucoma onset and progression involves circadian system deregulation. Several studies have shown the alteration of the melatonergic system in glaucomatous cases [2,3,4]. Melatonin is a circadian neurotransmitter produced in different districts of the body, including the eye, where it is involved in the control of daily physiological IOP fluctuations [5,6]. Excluding its intrinsic antioxidant activity, melatonin participates actively in the signaling pathway to decrease IOP in *Homo sapiens* by activating the human melatonin receptors 1 (*h*MT1) and 2 (*h*MT2) [7]. Thus, the clinical use of melatonin could help restore a physiological situation in glaucomatous eyes thanks to its antioxidant and signaling activities. However, this endogenous ligand has low chemical stability and bioavailability [8]. Therefore, we set out to find melatonin agonists with high melatonergic activity, possibly associated with high stability and low toxicity. We addressed, by an in silico approach, the binding modes of melatonin and one of its derivatives, agomelatine (Figure 1), within the orthosteric sites of *h*MT1 and *h*MT2 to evaluate the necessary molecular basis for agonist binding. Despite being a melatonin-derivative, agomelatine shows better chemical stability than its parent compound, and its crystallographic binding pose within the orthosteric site of *h*MT1 [9] can help retrieve much useful information to explore both human melatonergic target structures and the active binding poses for ligands to be tested [10]. Moreover, agomelatine is also used in antidepressant therapy for its atypical action that involves both the melatonergic and the serotoninergic pathways, through antagonist binding to the serotonin receptor 5HT2_C_ [11]. To validate the computational results, we used normotensive rats and rats with increased IOP to measure the effects of melatonin/agomelatine eye drops, comparing them with those drugs commonly used to reduce IOP in glaucoma patients. The hypotensive effect of melatonin and agomelatine on normotensive rats was evaluated as well.

## 2. Results and Discussion

### 2.1. Structural Features of hMT1 and hMT2

According to the current classification A–F [12,13] and ‘GRAFS’ [14] systems of amino acidic sequence similarity for G-protein coupled receptors (GPCR), melatonin receptors (MTs) belong to the class A or ‘rhodopsin-like family’, which in turn represent the largest GPCR superfamily [12,15]. More precisely, *h*MTs present a topology characterized by a counter-clockwise bundle of seven-pass transmembranes (TMs) and the association with guanine nucleotide-binding proteins (G proteins), which are two peculiarities of all the members belonging to the wide GPCRs superfamily [16].

Considering the protein structures in light of the inactive and active states of the β_2_-adrenergic receptor (prototype of GPCRs class A) [17,18], the available crystallographic data about MTs represent molecular complexes of the inactive receptor-binding melatonergic agonists [10,19]. Crystallized MTs binding melatonin-like activity ligands show an inactive conformation because they are induced by the presence of chimeric parts to thermo-stabilize the polypeptide chain during the crystallization process. Generally, an agonist binds better to the active form of the receptor than to the inactive one. Moreover, the literature indicates that the binding modes of the ligand and the residues of the orthosteric site of the GPCR are conserved during the transition from the inactive to the active form and vice versa [20]. Therefore, the available crystallized MTs could be used to search for new potential melatonergic agonists [20,21]. The built-up structures of *h*MT1 and *h*MT2 are in an inactive conformation since they conserve spatial atomic coordinates of the starting crystallographic structures of 6ME3 [10] and 6ME6 [19].

The built-up structures of *h*MTs present a hepta-helical counter-clockwise bundle with an extracellular N-ter and intracellular C-ter domain. TMs are linked by three extracellular loops (ECLs) and three intracellular loops (ICLs) (Figure 2). The *h*MTs present a further short amphipathic helix (helix-VIII) with a parallel orientation to the intracellular membrane side [10,19].

MTs participate in melatonergic signal transduction, activated by melatonin, its endogenous neurotransmitter, after binding in its orthosteric site. Crystallographic data on the spatial disposition of melatonin within MT pockets are not available yet. However, in PDB [23], different crystallized structures are available, matching the macromolecular complexes of MT1 and MT2 with known melatonergic agonists [10,19]. The superimposition of crystallized structures on the *h*MTs allows for more accurate identification of the ligand binding domain on human receptors. Both the *h*MT1 and *h*MT2 orthosteric sites include residues belonging to TMs 2,3,4,5,6,7, and ECL2 (Figure 3).

From a structural point of view, we found that *h*MTs orthosteric sites follow the same class A GPCRs superfamily topology, thus identifying *h*MTs clefts in an extracellular position of the receptor [10,19,24,25]. The assembly of all TMs, except TM1, constitutes the pocket sidewalls, and ECL2 the roof. The main differences between *h*MT1 and *h*MT2 protein domains are in the proximity of the roof, TM6 and ECL3 near ECL2, and sidewalls, with a final extremity of TM4. In *h*MT2, moving away the TM6 and ECL3 amino acidic backbone brings decreasing hydrophobic interactions with ECL2, maybe generating a further access route for the ligand^19^. This path seems to be impracticable in *h*MT1. On the other hand, *h*MT1 shows a route between the helixes TM4-TM5 for the access of ligands bigger than *h*MT2 since the roof of the pocket, ECL2, is bound more strongly to TM1, TM2, and TM7 [10] (Figure 4).

To point out other differences within *h*MTs binding sites, we further evaluated the whole-cavity volumes to identify changes in sub-pockets accommodating ligand functions or cleft access routes.

Through the CASTp server online [26], we calculated cavity volumes for both orthosteric sites of *h*MTs. Using the alpha-shape method [27], the CASTp server [28] draws bulbs to describe geometric and topological features for each cavity present in the protein structure. This server retrieved each potential cavity of *h*MTs structures, measuring volume and area [29,30,31]. In line with crystallographic data, the calculated size of the *h*MT2 cleft overcomes the *h*MT1 volume by about 54.0 Å^3^ since CASTp [26] calculated a cavity volume of 194.5 Å^3^ for *h*MT1 and 248.5 Å^3^ for *h*MT2. The main topological cavity differences were on the roof access route and in the sub-pocket accommodating indole C-2 substituents (Figure 5A). Different geometry of the *h*MT2 pocket in the roof provides a communication channel oriented outside, supporting a potential access route for the ligand as previously described. The sub-pocket accommodating substituent in the C-2 position of melatonin-derivatives indole developed itself toward an intracellular direction, enhancing the *h*MT2 pocket. Fewer changes involved sub-pockets where we found alkylamide and aromatic moieties of melatonin derivatives (Figure 4) [10,19].

The *h*MTs orthosteric sites present a similar amino acidic pattern close to 5 Å from molecular probes. Cleft constituent residues mainly involved in binding show conserved spatial positions in the site. Hence, we identified through UCSF Chimera [32] and PLIP server online [33] which residues and their respective intermolecular interactions are important for ligand binding.

All the molecular probes are melatonin analogs, sharing common pharmacophoric features. Each melatonin-like compound shares a similar molecular structure composed of a condensed aromatic scaffold bound to C-3 alkylamide and C-5 alkoxy or de-hydro furan functions. Additionally, in 2-melatonin derivatives, a further substituent on the C-2 position is also present. Following the previously described simplified structural scheme for each molecular probe, we considered molecular moieties to give an orthosteric site overview (Figure 5) (Appendix A).

The aromatic condensed system of each probe fits in a lipophilic sub-pocket of the protein site. Thus, aromatic core results flanked from TMs 3, 4, 5, and above have ECL2 as a roof. Both *h*MTs clefts present a prevalent hydrophobic nature for the presence of aliphatic side chains belonging to residues with conserved positions on protein domains. Mutagenesis studies showed the importance of F179^ECL2^ in *h*MT1 and F192^ECL2^ in *h*MT2 in the binding because isoleucine or alanine substitutions produced a loss of ligand affinity [10]. We noted the participation of all F^ECL2^ in a parallel π stacking intermolecular interaction with aromatic function to improve the global packaging of ligands in the pocket.

Even if the sub-pocket has lipophilic nature, it presents a conserved residue of N^4.60^ on TM4 of *h*MTs. The side chain of that residue act as a hydrogen bond donor towards loin pairs on the oxygen of ligand alkoxy or de-hydro furan functions. Despite its conservation in *h*MTs, the substitutions with alanine produce only *h*MT1 inactivation, while *h*MT2 retains its activity [10].

*h*MTs accommodate the ligand alkylamide function in a region of the orthosteric site defined by TMs 5 and 6 on sidewalls and ECL2 on the roof. Orientation of the alkylamide function assumes a ‘*tail down*’ or ‘*tail up*’ conformation towards Q181^ECL2^ of *h*MT1 or Q194^ECL2^ of *h*MT2, respectively. The better conformation in the *h*MT2 sub-cleft favors hydrogen bond formation between the ligand and glutamine side chain [10]. Some of the crystallized melatonin analogs present a further function on the C-2 position of the aromatic core, localized in a sub-pocket defined by TMs 2, 3, 5, and 6. Below the plane of the C-2 function, helix 6 presents a conserved CWXP motif, characteristic of GPCRs class A. In *h*MTs, we can see an amino acidic sequence CWAP which in turn has different residues employed in the binding and activation mechanism of the receptors [34,35,36,37,38,39,40,41,42]. Mutagenesis studies on residue P^6.50^, respectively P253^6.50^ in *h*MT1 and P266^6.50^ in *h*MT2, substituted with alanine show an unfavorable sub-pocket geometry that causes a complete loss of 2-iodomelatonin binding [42]. Thus, P^6.50^ results are important for the maintenance of the sub-cleft packaging. The CWXP motif presents a further conserved W^6.48^ that is involved in activating helixes movements. The coupling between the W^6.48^’ *rotamer toggle switch*’ and D^2.50^’ *allosteric switch*’ is at the base of the ‘*micro-switch*’ mechanism of the receptor. The activation of this mechanism triggers the contraction of the TM3–TM5–TM6 interface, and the roto-translation of TM6 triggers the subsequent activation of the TM3 to TM7 movement. All these protein changes bring the recruitment of G-protein on the intracellular side of the receptor [43].

### 2.2. Exploration of Agonists’ Binding Poses within MTs Orthosteric Sites

Taking advantage of the crystallographic data on the binding positions of melatonin analogs (agomelatine in 6ME5 [10] and 2-phenylmelatonin in 6ME6 [19]), we docked melatonin and its derivatives in the *h*MTs pockets. The similarities and differences in the binding modes between the two compounds and their affinity in terms of binding free energy (ΔG_bind_) were deeply analyzed.

Agomelatine in 6ME5 [10] constituted an excellent starting and reference point in analyzing the molecular docking results of this melatonin-like compound within the orthosteric site of *h*MT1.

It was very difficult to set up the docking parameters to reproduce the crystallized pose of agomelatine in *h*MT1 as each docked cluster showed an ensemble with a small number of conformations and a ΔG_bind_ range close to 1 Kcal/mol. This peculiar behavior must be ascribed to the high conformational flexibility of agomelatine together with the width (volume) of the *h*MT1 orthosteric site. The accurate cluster selection was achieved by crossing the docking results with the agomelatine conformational population and energy achieved at the Density Functional Theory (DFT) level using B3LYP hybrid functional and 6-311G* basis set (G09 software) [44]. As a result, we found out that the cluster corresponding to the crystallized pose corresponds to the lowest energy minimum for agomelatine (2.8 kcal/mol lower than that of the conformation present in the most populated cluster), and thus this pose proceeded with MD simulation.

The same protocol was applied to choose the best docked ligand positions for agomelatine within the *h*MT2 pocket (Figure 6).

Agomelatine shows a more favorable ΔG_bind_ than melatonin with both *h*MT1 and *h*MT2, thus suggesting a higher stability for the corresponding complex concerning the natural ligand (Table 1). Furthermore, the scientific literature supports the in silico findings since the ability of agomelatine to bind MTs is slightly better than the melatonin one [45,46]. In addition, agomelatine shows a slightly higher affinity for *h*MT2 than *h*MT1, and this data is also supported by experimental evidence, even if agomelatine is not considered a selective agonist of *h*MT2 [9,47].

### 2.3. Complexes Conformational Stability: Molecular Dynamics Simulations

As previously reported, applying a semi-flexible molecular docking protocol neglects the degrees of freedom of the biological molecular target, which only molecular dynamics could finalize, considering the whole system is flexible. Thus, starting from the selected docked conformations, we proceeded through MD stabilization to evaluate how the ligands’ binding modes in the *h*MTs orthosteric site can evolve over time. Analyzing the MD trajectories, Root Mean Square Deviation (RMSD), as a measure of the positional deviation of the atoms of a structure and its reference [48], was calculated. In addition, the RMSD evaluates whether the molecular dynamics trajectories have reached an equilibrium state and the quality of the process (Appendix A). In Figure 7, the protein RMSD as a function of simulation time is represented. Each of the simulated systems reaches the equilibrium state within the first 50 ns. The protein structure of *h*MT1 binding agomelatine changes more during the simulation time and requires more time to reach a final steady state.

The ligands’ RMSD was also calculated along the MD trajectory and is reported separately (Appendix A). In MT2, the RMSD vs. time plot of both agomelatine and melatonin shows a very similar trend. In MT1, agomelatine’s RMSD plot deviates prior to reaching the equilibrium state due to a rearrangement of the ligand’s conformation inside the cleft. This does not happen inside *h*MT2, in which the conformation of agomelatine changes less, suggesting a stronger binding affinity. The binding modes of melatonin in *h*MT1 and *h*MT2 remain similar to the starting poses. In Figure 8, the time evolution of H-bonding intermolecular interactions among ligands and receptors is reported along the MD trajectory.

Most hydrogen bonds are lost during the MD simulation trajectory. Except for the melatonin-*h*MT2 complex, residues Q181^ECL2^ for *h*MT1 and Q194^ECL2^ retain their hydrogen bonding (H-bonding) with the oxygen atom of the carbonyl group in the alkyl amide function of the ligands. The amide moiety of agomelatine acts as an H bond donor for the backbone carbonyl for N268^6.52^ in *h*MT2. The ligands’ methoxy group is an H bond donor for the amide group of N175^4.60/61^, the side chain in *h*MT2, but not for the corresponding residue N162^4.60/61^ in *h*MT1, which remains close to the methoxy function. Moreover, only in *h*MT1 are the carbonyl oxygen atom of the T178^ECL2^ side chain and the carboxyl group of G104^3.49^ fare involved in H bonding with the hydrogen atom attached to the melatonin indole nitrogen.

As previously anticipated, the aromatic scaffold is mainly stabilized by lipophilic interactions. According to MD simulations, F179^ECL2^ in *h*MT1 and F192^ECL2^ in *h*MT2 continue to bind the ligand. In particular, F^ECL2^ generates a π-π stacking interaction with the aromatic ring of melatonin in *h*MT1 and agomelatine in *h*MT2. In addition, the interaction of M107^3.32^ in *h*MT1 and M120^3.32^ in *h*MT2 with the aromatic scaffold of agomelatine and melatonin is observed, respectively, while F^ECL2^ in the proximity of the ligand is involved in van der Waals hydrophobic interactions with the aromatic ring or methoxy group of the ligand. According to MD simulations, melatonin increases the number of π- or Van der Waals interactions close to the methoxy group. The *h*MTs preserve the topology of residues V^5.42/43^ (V191 in *h*MT1, V204 in *h*MT2) and X^4.57^ (V159 in *h*MT1 and L172 in *h*MT2), residues used to generate alkyl interactions with the methoxy group. At the end of the MD simulation, agomelatine has a smaller number of π- or alkyl interactions stabilizing the naphthalene and methoxyl function with respect to the initial docked pose. All other residues listed in Figure 9 bind the ligand via weak van der Waals interactions. The number of these interactions is higher in the intermolecular complexes binding agomelatine than in those binding melatonin. This could be due to the different aromatic framework of the ligand. At the end of the process, five topological positions of residues involved in van der Waals interactions with the ligand are conserved in all four simulated systems, namely F^3.30^ (F105 in *h*MT1, F118 in *h*MT2), G^3.33^ (G108 in *h*MT1, G121 in *h*MT2), V^5.43/44^ (V192 in *h*MT1, V205 in *h*MT2), L^6.51^ (L254 in *h*MT1, L267 in *h*MT2), and Y^7.39/38^ (Y281 in *h*MT1, Y294 in *h*MT2).

In Figure 10, MMPBSA energy for each ligand-receptor molecular complex studied focuses on the last 10 ns of MD simulations, in the time frames in which RMSD plot reveal that all complexes reached a steady state. MMPBSA energy change in function of simulation time shows that melatonin is more stable in the orthosteric site of *h*MT1 than in *h*MT2. It is reversed for agomelatine in *h*MT2, where this derivative is more stable than the natural ligand. From the scientific literature, agomelatine has a better affinity for *h*MTs [9,49] than its natural ligand [9,50,51]. The time plot of MMPBSA energy for the *h*MT2 ligand seems to show the same trend of energy between melatonin and its derivative. The MMPBSA plot of the melatonin-*h*MT1 complex shows a lower free binding energy than agomelatine, suggesting that the natural ligand has the best affinity for the receptor.

In summary, the MMPBSA plot shows that melatonin has better stability in the *h*MT1 orthosteric site than in the *h*MT2 one, while the opposite is observed for agomelatine; overall, agomelatine shows a better affinity for hMTs [9,49] than the natural ligand, in line with the available experimental data [9,50,51].

### 2.4. IOP Effects In Vivo: Melatonin and Agomelatine Relative Efficacy

Melatonin 0.5% formulated in eye drops (see M&M) was given to rats, and its effects were compared with the hypotensive ability of commercially available drugs: timolol 0.5% and brimonidine 0.2%. These two agents are routinary used in the treatment of glaucoma because of their ability to lower the IOP, with brimonidine having selective α2-adrenergic agonistic activity [52], while timolol is a β-blocker [53]. Figure 11A shows the hypotensive effect in normotensive animals. Timolol and brimonidine had similar efficacy, lasting about 2 h and decreasing IOP at peak (15–30 min) by 33% of the initial value. Melatonin effects lasted a little longer, up until 3 h, with a deeper IOP lowering effect (47% decrease over control value), also seen at 15–30 min from instillation. Figure 11B reports the results obtained in the hypertensive rat model, obtained by clogging the trabecular meshwork with 2% MCE. The relative ratios among the three eye drop products remained similar. Again, timolol and brimonidine showed almost superimposable curves, peaking at 15–30 min at a 25% decrease, with an efficacy lasting over 6 h. The melatonin effect peaked at 30 min at a 40% decrease, with the efficacy also lasting over 6 h. Figure 12 shows the comparison in the normotensive rat between the hypotensive effects of melatonin and agomelatine eye drops, both formulated at 0.2% (see M&M). A lower concentration was chosen in this case to show a dose-dependent effect in potency and duration. Both compounds peaked at 15 min at around a 35% decrease. Likely due to the lower concentration, melatonin effects lasted only one hour, while agomelatine efficacy was protracted over 3 h.

Thus, agomelatine was found to have a similar potency to melatonin but a longer-lasting efficacy in lowering the IOP. The same potency can be ascribed to a comparable affinity for the two *h*MTs receptors with respect to melatonin, as the computational data found; besides, the long-lasting effect could be instead attributed to the involvement of the serotonin receptor 5-HT2c, which is not bound by the natural ligand melatonin. Further studies are currently underway to confirm this hypothesis.

### 2.5. Searching for Potential Melatonin Agonists: Test Case Studies

Using all the structural information obtained from the in silico results and considering that agomelatine inhibits the 5-HT2C receptor besides its melatonergic effect, we performed a virtual screening of known antagonists/inverse agonists for this receptor towards MT1 and MT2 using Autodock Vina. Finally, we chose some promising compounds that bound efficiently into the orthosteric cavity. In Table 2, the binding affinities of three selected potential agonists are reported.

## 3. Material and Methods

### 3.1. D Protein Modeling

Human melatonin receptors (*h*MTs) crystallographic structures were obtained from the *Brookhaven Protein Data Bank* (www.prcsb.org/pdb) [23] [pdb codes 6ME3 for *h*MT1 and 6ME6 for *h*MT2, both in complex with a melatonin agonist, 2-phenylmelatonin]. The retrieved 3D structures were aligned with the FASTA sequences reported in UniProt [54] P48039 (*h*MT1) and P49286 (*h*MT2) to identify any aminoacidic differences or lacking residues.

In both structures, engineered chimeric proteins bonded to the receptors used for the crystallization process were present; therefore, they were firstly removed, then substituted with the receptor residues according to UniProt *h*MTs sequences [54] to obtain the full *h*MTs 3D structures, i.e., P23^1.28^-V318^C-ter^ of the sequence P48039 for *h*MT1 and W38^1.30^-N328^VIII-helix^ of the sequence P49286 for *h*MT2. Both structures present a disulfide bridge between C^3.25/25^-C^ECL2^, respectively C100^3.25/25^-C177^ECL2^ in *h*MT1 and C113^3.25/25^-C190^ECL2^ in *h*MT2. After adding hydrogen atoms using CHIMERA [32], each structure has been submitted to OPM-Orientations of Proteins in Membranes database [55] to get the *h*MTs spatial distribution in the biological membrane through comparison with the spatial disposition of known proteins characterized by similar structure, classification topology, and cellular localization in relation to the analyzed structure [56]. The whole protein structure of each *h*MTs has been aligned and cross-compared to investigate the orthosteric site composition to better elucidate the aminoacidic interactions with the ligands.

Key residues involved in ligand recognition, molecular docking, and molecular dynamics have been highlighted through UCSF Chimera [32], PLIP server online [33], and Discovery Studio [57]. After a detailed evaluation of the pockets’ aminoacidic composition, their volume was calculated with the CASTp server [26], which used the alpha shape method [27] to describe geometric and topological features for each cavity and their own volume and area [28,29,30,31]. Finally, the macromolecular receptors were minimized and stabilized in their biological environment prior to proceeding with the docking calculations.

### 3.2. hMTs Ligands Docking

Ligands were prepared using UCSF CHIMERA software^32^ (i.e., melatonin, the natural ligand of melatonin receptors (MTs), and agomelatine, a melatonin-mimetic compound exerting activity like the natural ligand) [9,58] (Figure 1).

An in silico dynamic docking protocol was set to compare the ligands’ binding poses in the *h*MTs orthosteric sites. Dynamic docking combines molecular docking with atomistic molecular dynamics simulation to improve the exploration of the mechanism and free binding energy (ΔG_bind_) behind ligand-protein interaction [59]. Initially, via semi-flexible molecular docking, the best binding modes between two molecular partners [60] were predicted in the *h*MTs orthosteric site. Autodock4.0 software [61] was used concerning *h*MTs with fixed atomic spatial coordinates and ligand flexibility to explore its conformational space, including all ligands’ torsional degrees of freedom. The previous knowledge regarding the position of the *h*MTs orthosteric sites allowed the execution of a focused docking on these pockets, setting up the grid box size: for *h*MT1: 70 × 68 × 66 Å^3^ and *h*MT2: 76 × 70 × 58 Å^3^.

The genetic algorithm (GA) setting the runs to 100 and the Lamarckian genetic algorithm (LGA) for data analysis were used to identify the best scored binding poses [62]. The best poses were selected according to their lowest ΔG_bind_ values (high affinity). Since the crystallographic data about *h*MTs with the natural ligand are not available in PDB [23], we compared melatonin with the agomelatine in complex with *h*MT1 docking poses in 6ME5 [10] and 2-phenilmelatonin in complex with *h*MT2 in 6ME6 [19].

The identified docked ligand-protein complexes were further repositioned in orthosteric sites of *h*MT1 and *h*MT2 using a focused docking approach to generate the final complex, that has then minimized prior to proceeding to the subsequent molecular dynamics simulations. All six molecular complexes were then oriented in a lipid bilayer as *h*MTs using the OPM server database [55].

### 3.3. Molecular Dynamics Simulation of the hMTs-Ligands Complexes in Membrane

Molecular dynamics (MD) simulations in a membrane at physiological conditions were carried out for the *h*MTs-ligand complexes using GROMACS 2020.6 (https://manual.gromacs.org/) [62] in order to allow the receptor and its ligand fitting (Figure 13). The MD workflow provides three main steps: system preparation, simulation, and trajectory analysis. Melatonin and agomelatine charges were calculated with AM1/BCC method [30]. Ligands’ parameters were retrieved with parmchk2 after the conversion of GAFF (General AMBER Force Field) [31] to AMBER ff14 atom types via the AMBER LEaP program, using AMBERff14SB [63] force field Topology; the ligands structural files were then converted from AMBER to GROMACS input files [58,59]. The assembled molecular systems (*h*MTs alone and in complex with ligands in the membrane) were retrieved by the CHARMM-GUI membrane builder (www.charmm-gui.org) [64,65,66]. All the *h*MTs were put into a lipid bilayer composed of phosphatidylcholine (POPC) and cholesterol (CHL1) using the spatial distribution of the OPM database [55,67].

Water and ions were added to reach the physiological concentration of 0.15 M, using sodium (Na^+^) and chloride (Cl^−^) to neutralize the charge of the macromolecular system and TIP3P model for the solvent. AMBERff14SB force field [68] was used for energy calculation within Periodic Boundary Conditions (PBC) to decrease edge effects of tangential boxes in which the analyzed system is set; box size and total atomic number are reported in Appendix A.

Van der Waals forces calculations were carried out with a double cut-off value: 10–12 Å. Particle-Mesh-Ewald method allowed calculation of long-range electrostatic forces [69]. LINear Constraint Solver (LINCS) algorithm for hydrogen atoms was used with 2 fs as the time step [70]. All molecular systems underwent a complete minimization protocol prior to proceeding with MD simulations [71]. The equilibration was achieved starting from an initial canonical ensemble (NVT) equilibration step using temperature control of 310 K with a Berendsen thermostat [72] followed by three other following equilibration steps in an isobaric-isothermal ensemble (NPT). In the NPT ensemble, we maintained semi-isotropic conditions, keeping a constant 1 atm pressure. At the end of the process, a molecular dynamics simulation of 100 ns on each system was run. The temperature and the pressure of this process were controlled with a Nosé–Hoover thermostat [73,74] and a Parrinello–Rahman barostat [75].

The molecular mechanical Poisson-Boltzmann surface area (MMPBSA) calculation was carried out to estimate the free Gibs binding energy of the simulated complexes [52]. MMPBSA methods are the most commonly used approaches to estimate the protein–ligand binding affinities. This calculation method provides an average of the free binding energy measured for a conformational ensemble of ligand-protein complex conformations [76,77].

### 3.4. In Vivo IOP Reduction Effects of Melatonin and Agomelatine

Animals were used in agreement with the Association for Research in Vision and Ophthalmology statement for the Use of Animals in Ophthalmic and Vision Research. The study also agrees with the European Communities Council Directive (2010/63/UE) and the Italian guidelines for animal care (DL 26/14). The experimental protocol was approved by the Commission for Animal Wellbeing of the University of Pisa (protocol n. 133/2019-PR). Rats (Sprague Dawley strain, 200 g body weight, 2–3 months of age) were obtained from Charles River Laboratories Italy (Calco, Italy). Before handling for tonometry, rats were acclimatized for 1 week. To evaluate the effects of melatonin (given at 0.5% or 0.2%), 0.2% agomelatine and commercially available drugs commonly used to reduce IOP in glaucoma patients (i.e., 0.5% timolol and 0.2% brimonidine) on normotensive rats, 3 rats (6 eyes, 3 measurements/eye/time point) were used for each formulation. Rats were treated with 10 µL per eye of formulations of melatonin prepared in Soluplus 1 mM (Merck) and borate buffer, pH 7.4. Ten µL per eye of hypotensive drugs were administered as well. The IOP was measured from time 0 (before administration) to 6 h after administration using the TonoLab device (Icare, Finland).

For the hypertensive model, 3 rats (6 eyes, 3 measurements/eye/time point) were used for each formulation. Ocular hypertension was obtained through the injection in the anterior chamber of the rat eye of 15 µL of 2% methylcellulose (MCE), as previously described [78]. Twenty-four hours after the MCE injection, the rats were utilized to evaluate the effects of 0.5% melatonin or other hypotensive drugs (0.5% timolol and 0.2% brimonidine).

### 3.5. In Silico Screening for Melatonin Agonists

In silico screening towards hMT1 and hMT2 receptors was carried out using as molecular libraries all the known 5-HT2C antagonists and inverse agonists that were retrieved from the IUPHAR database (https://www.guidetopharmacology.org/) and Drug Bank repository (https://go.drugbank.com/). All molecular docking experiments were performed by AutoDock Vina version 1.1.2 [79,80]. The virtual screening software VINA was used to screen the collections of chosen drugs concerning the grid potential spread all over the protein surface. The resulting docked poses were clustered according to the protein regions in which the ligands bind. Finally, the resulting clusters were ranked based on the predicted binding energy and their population. Only clusters, including more than 10 compounds, were considered. Finally, the best scored compounds (top 100) were further selected based on their binding pose (compared with melatonin/agomelatine) in the orthosteric site and their specific interactions with cleft residues.

## 4. Conclusions

Glaucoma is one of the most important ocular neuropathies, often associated with an increased IOP. In this study, we aimed to open the way for the rational design of new compounds with high melatonergic activity and low toxicity by investigating the molecular basis of MTs agonists’ activity. As a result, the computational findings defined the structural basis for agonistic activity and selectivity toward human melatonin receptors MT1 and MT2 of agomelatine and melatonin.

Specifically, the binding poses of melatonin and its agonist agomelatine within the orthosteric sites of *h*MTs were studied by molecular dynamics/molecular docking protocols. We predicted the binding poses of the natural ligand and the agomelatine positioning both in *h*MT1 and *h*MT2 in the membrane. Moreover, the binding stability was studied in terms of free Gibbs binding energy. The intermolecular interaction patterns were defined for each ligand and the involved melatonergic receptors, considering also the dynamical evolution in the biological environment (i.e., membrane). The results indicate that both ligands interact with the same key residues. MMPBSA calculation predicts that the two ligands have almost the same affinity towards the two *h*MTs receptors, as confirmed by experimental results. However, agomelatine shows a higher affinity for hMT2 than melatonin.

Besides, in vivo findings showed that agomelatine achieves a longer-lasting effect in downregulating IOP. This result could be explained by considering its additional known effect as an antagonist of the 5-HT2c serotonin receptor, which is also involved in the IOP regulation pathway [53]. Hence, this peculiar behavior can open new frontiers in finding novel melatonergic agonists with more durable long-term effects.

In fact, libraries of 5-HT2c antagonists have been collected and screened computationally regarding MT1 and MT2 to identify novel agonists. Among them, three compounds were selected based on their positioning inside the orthosteric site, binding pattern, and affinity, which were compared with melatonin and its agonist agomelatine. At present, in our laboratory, in silico and in vitro testing studies are ongoing, aiming to identify safe compounds ready to use in clinical practice within a drug repurposing strategy.

## Figures and Tables

**Figure 1 ijms-24-02863-f001:**
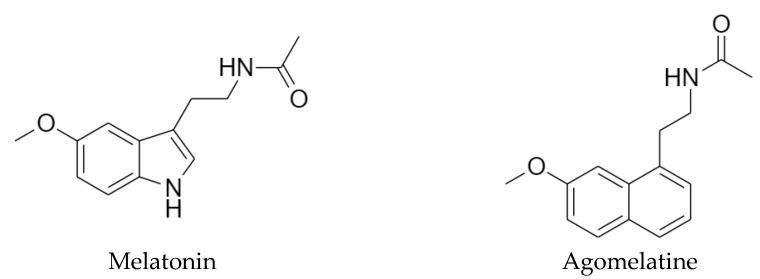
2D structure of melatonin and its agonist agomelatine.

**Figure 2 ijms-24-02863-f002:**
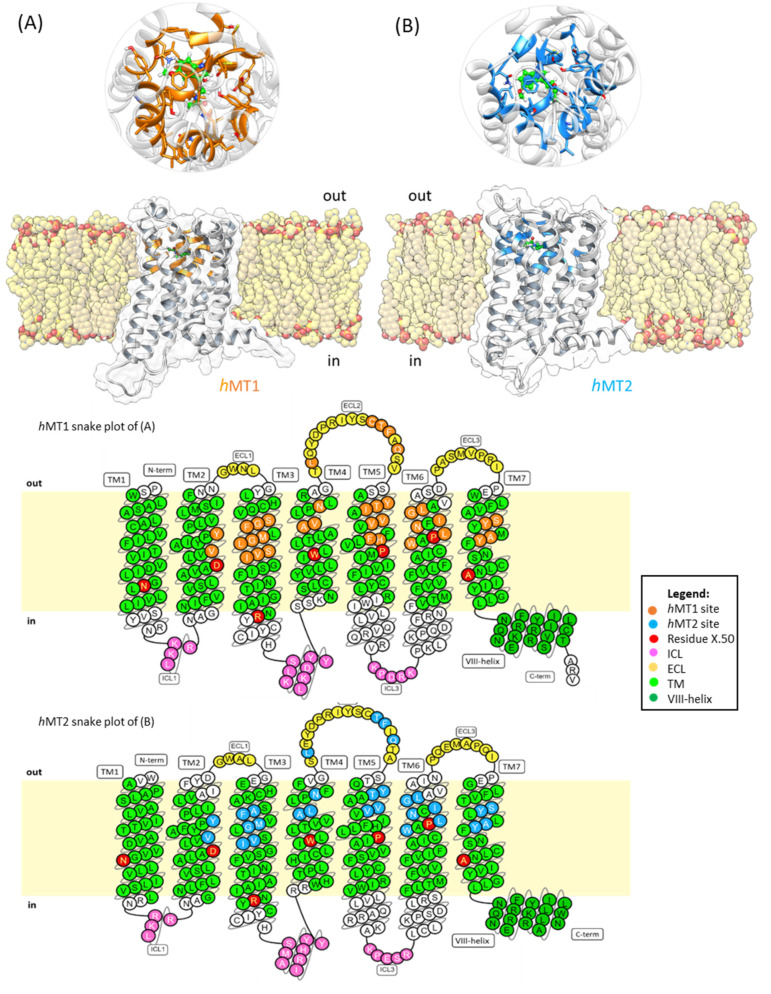
Snake plots reached by GPCRdb of the *h*MTs membrane topology, namely (**A**) *h*MT1 and (**B**) *h*MT2. Each TM has its own X.50 residues (red), references for Ballesteros–Weinstein numeration, and amino acids of the membrane (light green), repositioning the proteins with OPM sever. Also present are ICL (pink), ECL (yellow), and VIII-helix (dark green). The ligand (melatonin) is reported in green sticks. (**C**) The alignment shows the topology of the residues of the orthosteric sites of *h*MTs. Additionally, the snake graphs illustrate the residue distribution on the protein domains, associating the Ballesteros–Weinstein numeration to each residue of the clefts [21,22].

**Figure 3 ijms-24-02863-f003:**
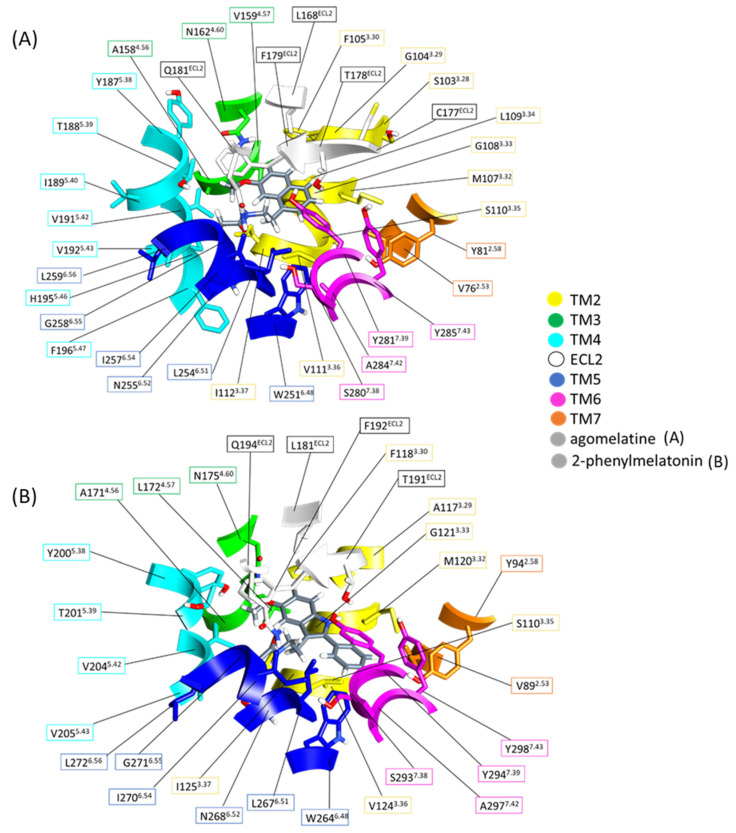
Residues at the orthosteric site of hMT1 (**A**) and hMT2 (**B**), binding agomelatine (with the same pose in 6ME5) and 2-phenylmelatonin (with the same pose in 6ME6). Each color indicates the receptor domain. TM: transmembrane domain, ECL: extracellular loop.

**Figure 4 ijms-24-02863-f004:**
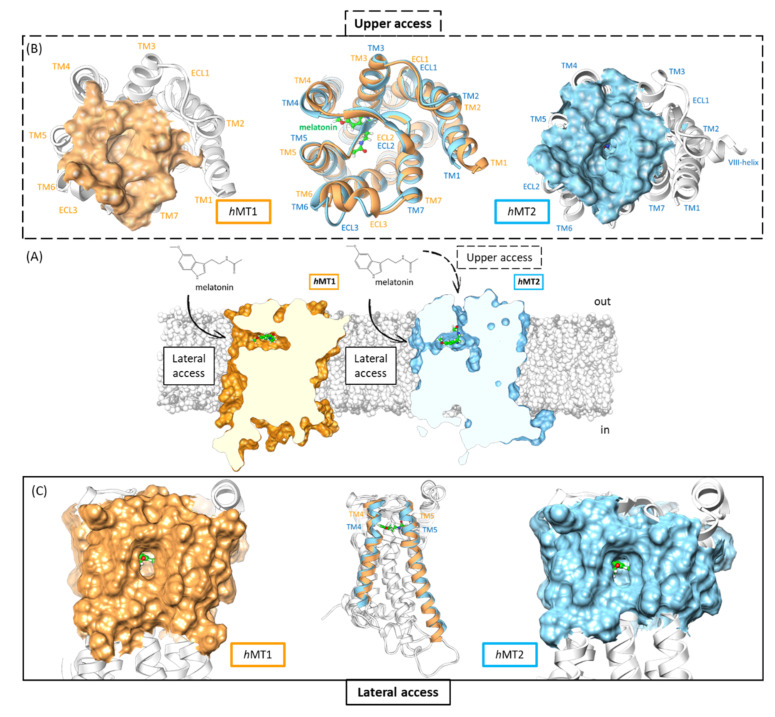
The *h*MT1 (orange) and *h*MT2 (light blue) structures. Melatonin is always in green. (**A**) depicts all potential ligand access routes for *h*MTs orthosteric sites. (**B**) represents a comparison between *h*MT’s top-side paths. (**C**) shows the overall superimposition of both *h*MT1 and *h*MT2.

**Figure 5 ijms-24-02863-f005:**
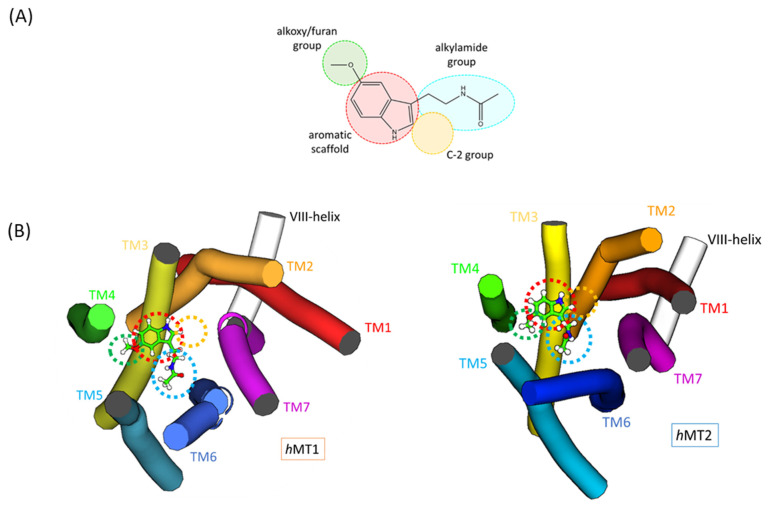
(**A**) pharmacophoric representation of melatonin analogs; (**B**) Spatial representation of disposition of pharmacophoric features in the 3D orthosteric site.

**Figure 6 ijms-24-02863-f006:**
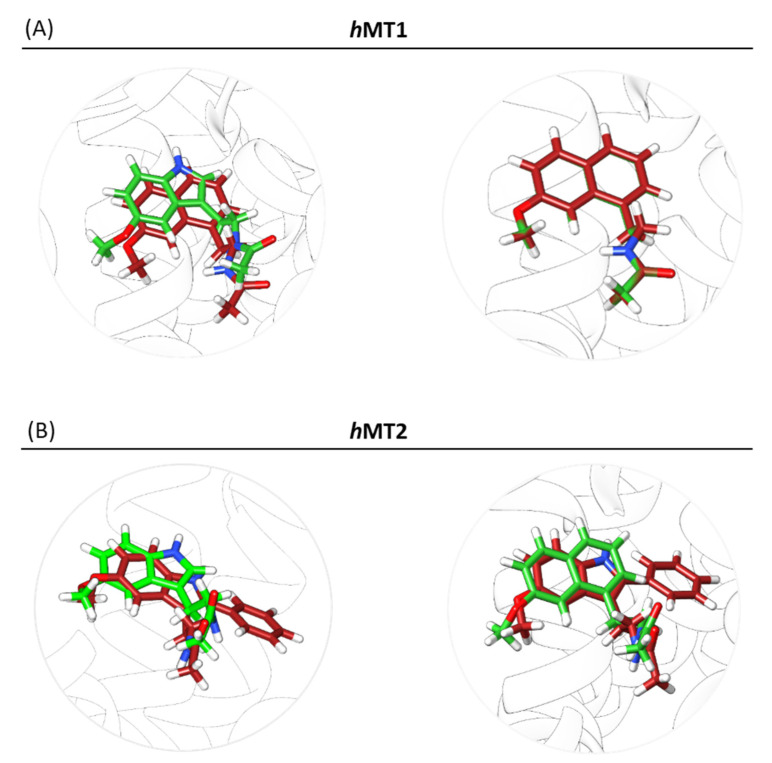
Docking poses of melatonin and agomelatine in hMT1 and hMT2 orthosteric sites. As a reference, the crystallographic conformation of agomelatine in 6ME5] for hMT1 (**A**) and melatonin in 6ME9] for hMT2 (**B**) are reported in red. The tested docked ligands are highlighted in green (melatonin on the left side, agomelatine on the right in both A,B sections).

**Figure 7 ijms-24-02863-f007:**
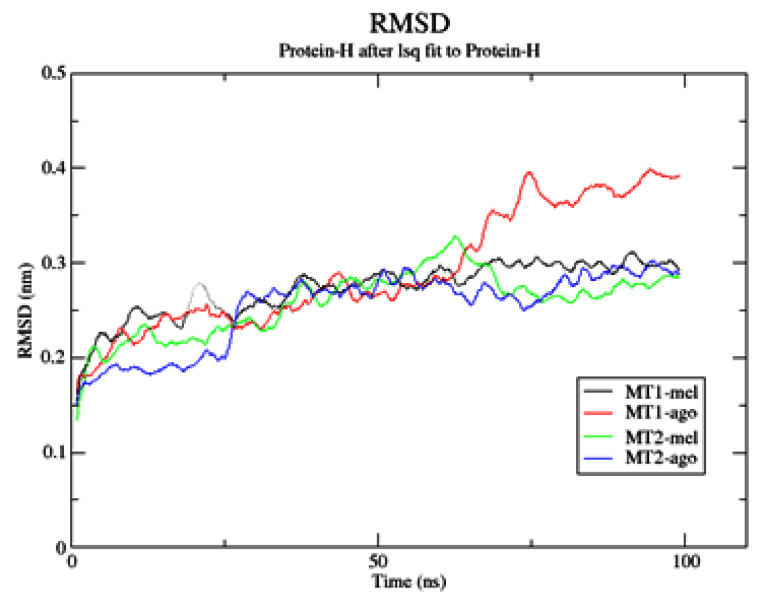
RMSD plot of *h*MTs protein backbone in binding complex with melatonin or agomelatine, in the function of time.

**Figure 8 ijms-24-02863-f008:**
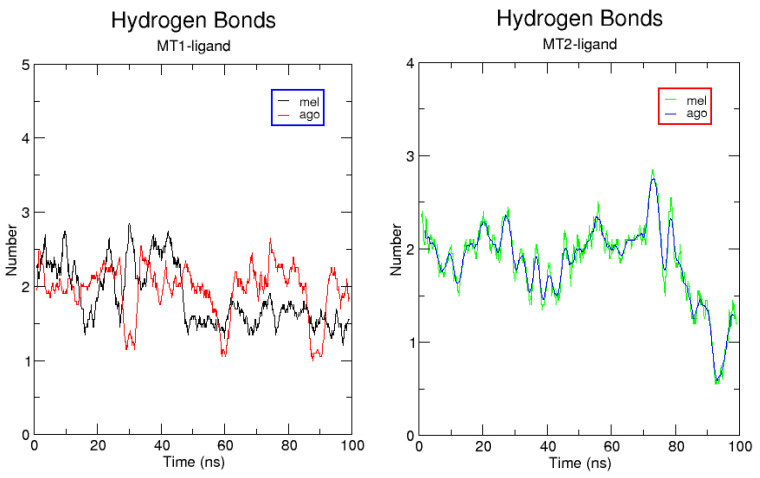
H-bonding interaction evolution along the 100 ns MD trajectory.

**Figure 9 ijms-24-02863-f009:**
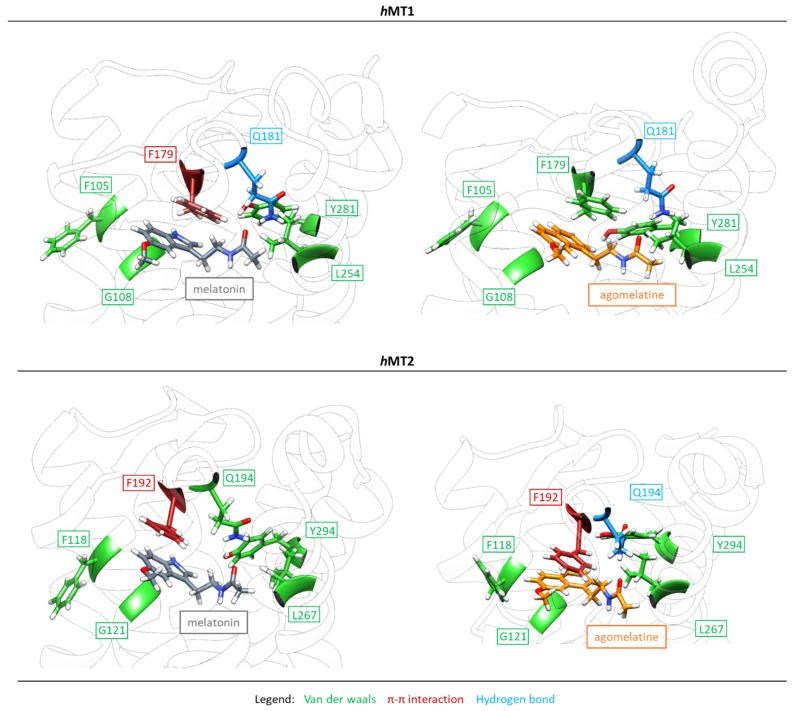
3D representation of the final poses of melatonin and agomelatine in the *h*MT1 and *h*MT2 orthosteric site after 100 ns molecular dynamics simulations (docked poses starting for MD in green; final MD configuration in yellow) (for 2D mapping see Appendix A).

**Figure 10 ijms-24-02863-f010:**
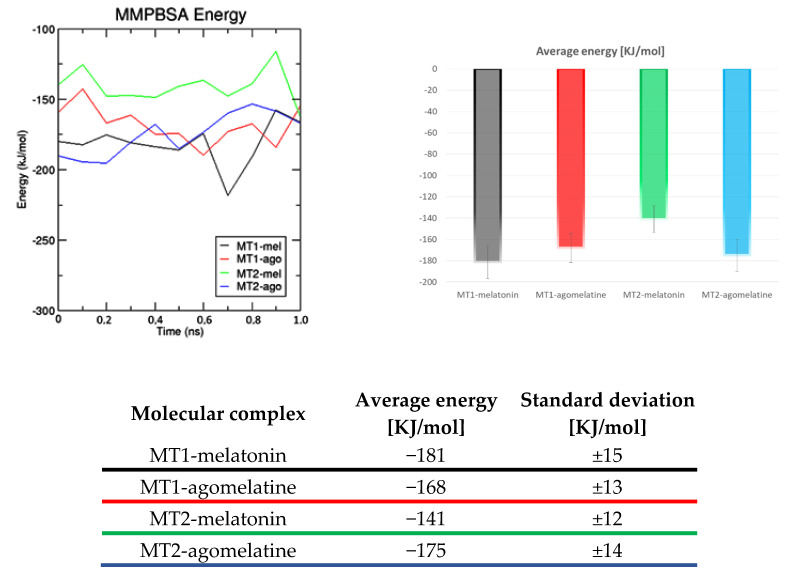
Melatonin and agomelatine affinity values towards *h*MTs in terms of free binding energy calculated using MMPBSA in 1 ns and molecular dynamics simulations.

**Figure 11 ijms-24-02863-f011:**
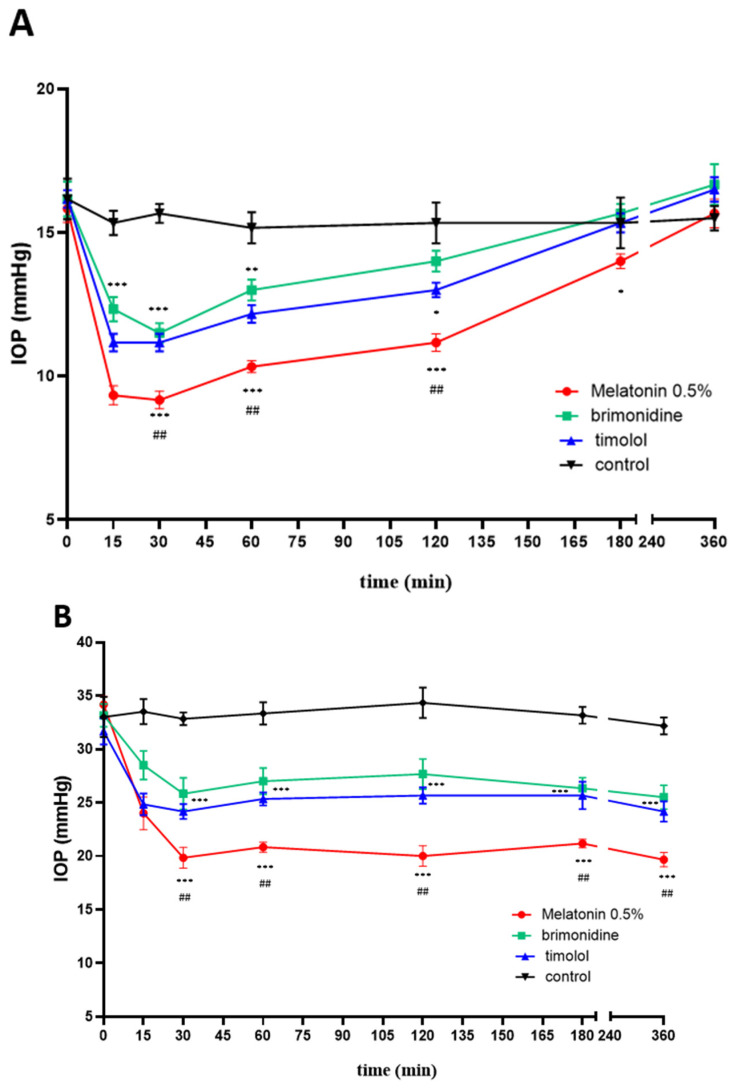
Time-course of the hyponotizing effects of melatonin 0.5%, brimonidine 0.2%, and timolol 0.5% in normotensive (**A**) or hypertensive (**B**) rats. *** *p* < 0.001, ** *p* < 0.01, * *p* < 0.05 vs. control; ## *p* < 0.01 vs. timolol or brimonidine.

**Figure 12 ijms-24-02863-f012:**
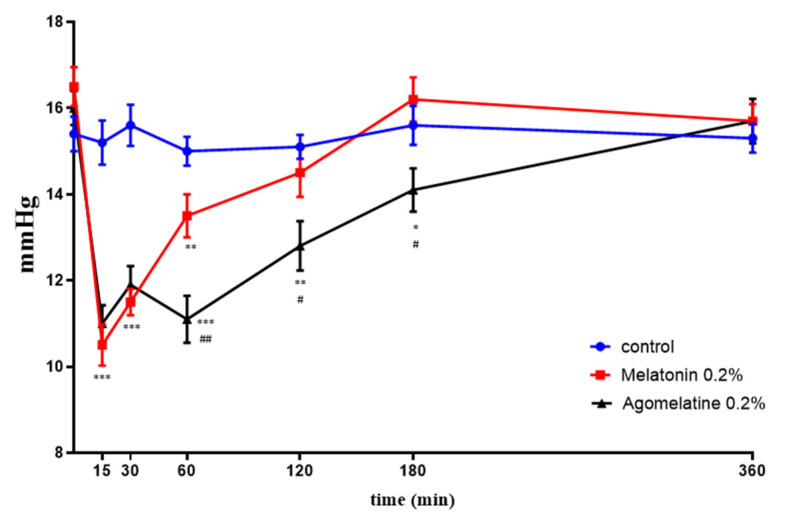
Time-course of the hypotonizing effects of melatonin 0.2% vs. agomelatine 0.2% in normotensive rats. *** *p* < 0.001, ** *p* < 0.01, * *p* < 0.05 vs. control; ## *p* < 0.01, # *p* < 0.05 vs. melatonin.

**Figure 13 ijms-24-02863-f013:**
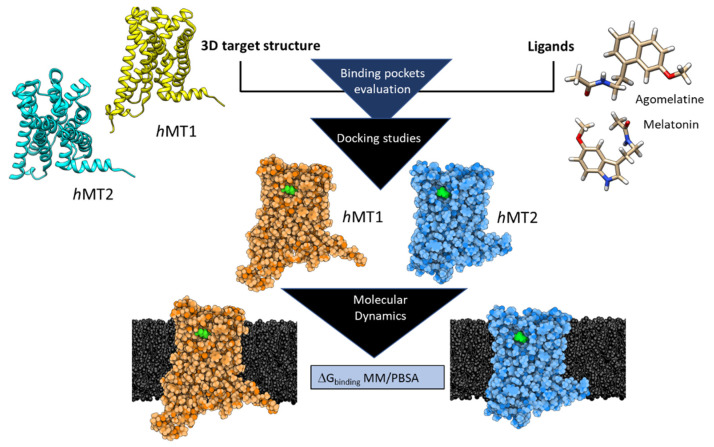
Computational Workflow.

**Table 1 ijms-24-02863-t001:** Free Gibbs binding energies (ΔG_bind_) associated with the docked poses are reported in Figure 6.

Receptor	Compound	Cluster Conformation	ΔG_bind_ (Kcal/mol)
*h*MT1	Melatonin	1 of 1	−6.42
	Agomelatine	2 of 7	−6.81
*h*MT2	Melatonin	1 of 1	−7.12
	Agomelatine	1 of 1	−7.52

**Table 2 ijms-24-02863-t002:** Binding energies of best-scored compounds compared to melatonin and its agonist agomelatine one.

Receptor	Ligand	Molecular Docking ΔG_bind_	Receptor	Ligand	Molecular Docking ΔG_bind_
*h*MT1	Melatonin	−6.42 Kcal/mol	*h*MT2	Melatonin	−7.12 Kcal/mol
	Agomelatine	−6.81 Kcal/mol		Agomelatine	−7.52 Kcal/mol
	Ramelteon	−7.93 Kcal/mol		Ramelteon	−8.66 Kcal/mol
	Clozapine	−8.47 Kcal/mol		Clozapine	−8.33 Kcal/mol
	Flumazenil	−7.43 Kcal/mol		Flumazenil	−7.71 Kcal/mol

## Data Availability

Data are also reported in the Appendix A.

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
