# Peer review of "Structural Basis for Agonistic Activity and Selectivity toward Melatonin Receptors *h*MT1 and *h*MT2"

_ijms, 2023, doi:10.3390/ijms24032863_

Round 1
Reviewer 1 Report
In this manuscript, Cantarini et al. implement a computational approach to investigate the binding characteristics of melatonin and its analogue, agomelatin, to their targets (MT1, MT2). They additionally study the IOP-lowering effect of both molecules. They report that melatonin and agomelatine are of comparable affinity towards their receptors and of similar potency, but agomelatine has a longer lasting IOP-lowering effect. The manuscript is well-written and the topic is of clinical relevance.
It is unclear to me why the authors compare melatonin given at 0.5% concentration with commercially available IOP-lowering drops, but then switch to 0.2% when comparing melatonin with agomelatine. Could the authors explain the rationale behind these choices?
Author Response
Thanks for the comments. We better specified in the text; the main reason is that a lower concentration was chosen to show a dose-dependent effect both in terms of potency and duration.

Reviewer 2 Report
In this manuscript, Galeazzi et al. has attempted to decipher the binding of melatonin towards its natural macromolecular targets using various computational approaches. Collectively, the study is well conducted and the manuscript language is scientifically sound. However, I would ask the authors to address the following comments to further improve the overall quality of the manuscript.
1. Authors may provide a workflow of the methodology or schema of the study.
2. As the aim of the study was to pave the way for the design of new compounds, a test case study on a set of compounds might have been conducted to actually show how the prediction of the binding poses for the hMT1 and hMT2 using melatonin and agomelatine is useful, at least computationally.
3. Conclusions might be modified to give more clear picture on the results obtained from the study.
Author Response
- Authors may provide a workflow of the methodology or schema of the study.
Answer: Thanks for the suggestion. We added Fig.13 at the end of the methods section, representing the computational workflow of the study.
- As the aim of the study was to pave the way for the design of new compounds, a test case study on a set of compounds might have been conducted to actually show how the prediction of the binding poses for the hMT1 and hMT2 using melatonin and agomelatine is useful, at least computationally.
Answer: The in silico/in vivo studies are still on going in our laboratory, but according to your suggestion, we added some preliminary computational data in a supplementary section (2.5 at page 16 and added also the correspondent method section 3.5).
- Conclusions might be modified to give more clear picture on the results obtained from the study.
Answer: Thanks for your suggestions. We re-elaborated the conclusion to make them more clear and representative of the study.
